# Effect of Grain Structure and Quenching Rate on the Susceptibility to Exfoliation Corrosion in 7085 Alloy

**DOI:** 10.3390/ma16175934

**Published:** 2023-08-30

**Authors:** Puli Cao, Chengbo Li, Daibo Zhu, Cai Zhao, Bo Xiao, Guilan Xie

**Affiliations:** 1School of Mechanical Engineering and Mechanics, Xiangtan University, Xiangtan 411105, China; xtucpl@126.com (P.C.); zcing0106@163.com (C.Z.); m17369283823@163.com (B.X.); xieguilan@xtu.edu.cn (G.X.); 2Engineering Research Center of the Ministry of Education for Complex Trajectory Processing Technology and Equipment, Xiangtan University, Xiangtan 411105, China; 3Guangdong Xingfa Aluminium Co., Ltd., Foshan 528137, China

**Keywords:** 7085 alloy, grain structure, cooling rate, quenching-induced phase, exfoliation corrosion

## Abstract

The influence of grain structure and quenching rates on the exfoliation corrosion (EXCO) susceptibility of 7085 alloy was studied using immersion tests, optical microscopy (OM), scanning electron microscopy (SEM), electron backscatter diffraction (EBSD), and scanning transmission electron microscopy (STEM). The results show that as the cooling rate decreases from 1048 °C/min to 129 °C/min; the size of grain boundary precipitates (GBPs); the width of precipitate-free zones (PFZ); and the content of Zn, Mg, and Cu in GBPs rise, leading to an increase in EXCO depth and consequently higher EXCO susceptibility. Meanwhile, there is a linear relationship between the average corrosion depth and the logarithm of the cooling rate. Corrosion cracks initiate at the grain boundaries (GBs) and primarily propagate along the HAGBs. In the bar grain (BG) sample at lower cooling rates, crack propagation along the sub-grain boundaries (SGBs) was observed. Compared to equiaxed grain (EG) samples, the elongated grain samples exhibit larger GBPs, a wider PFZ, and minor compositional differences in the GBPs, resulting in higher EXCO susceptibility.

## 1. Introduction

7XXX-series aluminum alloys are characterized by high strength and low density, making them widely utilized in industries such as automotive manufacturing and aerospace. However, this series of alloys exhibit a tendency for localized corrosion in corrosive environments, which encompasses phenomena such as pitting, intergranular corrosion (IGC), EXCO, and stress corrosion cracking (SCC) [1,2,3,4]. Corrosion issues lead to a decline in material performance [5], shortened service life, and restrict the wide application of this alloy series, making it a pressing problem to be addressed. Extensive research has been conducted by researchers to enhance the corrosion resistance of this alloy series.

The EXCO susceptibility of Al-Zn-Mg-Cu alloys is significantly affected by grain size and shape. Masoud et al. [6] demonstrated that high-angle grain boundaries (HAGBs) in Al-Zn-Mg-Cu alloys serve as relatively strong hydrogen trapping sites, possessing higher hydrogen desorption energies. This propensity results in the easy aggregation and retention of hydrogen atoms within HAGBs. Simultaneously, statistically stored dislocations exhibit moderate hydrogen desorption energies, continuously supplying hydrogen atoms to HAGBs, consequently leading to a sustained increase in hydrogen content within HAGBs. As a consequence, HAGBs tend to become the initiation points for corrosion and the propagation paths for crack propagation, thereby facilitating intergranular fracture. Wloka et al. [7] investigated the influence of recrystallized coarse grains on the EXCO susceptibility of AA7010 and AA7349 alloys. The findings indicated that the coarse grain layer increased resistance to EXCO. Huang et al. [8] discovered that in AA7075 and AA7178 alloys, the grain size and aspect ratio of grains are crucial factors in the kinetics of localized corrosion growth. For Al-Zn-Mg-Cu alloys, high aspect ratio grains generally exhibit higher EXCO susceptibility [9], as the accumulation of corrosion products during immersion in corrosive solutions leads to wedging stresses at GBs, resulting in elongated grain bubbling and a layered morphology. Equiaxed grains show intergranular corrosion but have low EXCO susceptibility [10,11,12]. For thick plates of Al-Zn-Mg-Cu alloys, during the solution and quenching processes, the grain structure often exhibits non-uniformity [13,14], with the cooling rate in the central layer being slower compared to that in the surface layer. As a result, quenching precipitates occur in the central layer and tend to reduce the mechanical and localized corrosion resistance properties [15,16]. For 7XXX alloys, the location of quenching precipitates varies with different grain structures, leading to varying losses in corrosion resistance and mechanical properties after subsequent aging [16,17].

Numerous studies indicate that the quenching rate has a significant influence on the EXCO resistance of 7XXX alloys. Sánchez-Amaya [18] found that slow quenching significantly enhanced the IGC susceptibility of the AA7075 alloy. Marlaud et al. [19] found that the resistance to EXCO of the 7449-T7651 alloy decreased with a decline in quenching rate. Li et al. [20] found that the EXCO grade of an Al-5Zn-3Mg-Cu alloy sheet gradually changed from P grade to ED grade as the quenching rate reduced from 2160 ℃/min to 100 ℃/min. Liu et al. [21] conducted exfoliation corrosion tests on 7055 alloy using end-quenching and found that the maximum corrosion depth of EXCO increased as the quenching rate decreased. Chen et al. [22] performed natural aging and artificial aging (T6) on a 7XXX-series aluminum alloy after end-quenching and compared their IGC results. They observed that the maximum corrosion depth of both aging samples increased with a decline in quenching rate, with the naturally aged sample exhibiting greater corrosion depth than the artificially aged sample. For 7XXX alloys, the characteristics of the quenched precipitates and the width of the PFZ are determining factors for the corrosion resistance of 7XXX alloys. The quenching rate affects the EXCO resistance of the 7XXX alloy by influencing the quantity, size, composition, and distribution of quenched precipitates, as well as the width of the PFZ [23,24,25,26].

The 7085 alloy holds a crucial position in the aerospace and defense industries, and its resistance to EXCO is a decisive factor in ensuring the long-term and safe utilization of the alloy. However, there has been limited research on the corrosion resistance of the 7085 alloy, and the influence of quenching-induced precipitation on the sensitivity to EXCO in the 7085 alloy with different grain structures is not yet clear; thus, its mechanism requires further investigation. This study simultaneously investigates the effects and mechanisms of grain structure and quenching rates on the EXCO sensitivity of the 7085 alloy, as well as the mechanisms of corrosion crack propagation.

## 2. Experimental

### 2.1. Materials

The experimental materials were homogenized ingots and hot-rolled thick plates of 7085 alloy. The hot-rolled deformation of the plates was 85%. The chemical composition is shown in Table 1. The end-quenching samples with dimensions of 25 × 25 mm and a length of 125 mm were cut from the homogenized ingots and hot-rolled plates. At one end of the samples, a circular groove with dimensions of 22 mm in diameter and 10 mm in depth was processed to serve as the water spray end. At the other end, a threaded hole with dimensions of 5 mm in diameter and 15 mm in depth was created to secure the samples on the sample holder for end-quenching. The specimens underwent solution treatment by heating in an SX-4-10 resistance furnace to 470 °C, followed by a holding time of 1 h, and then were rapidly removed for water quenching at the groove end (transfer time less than 15 s). The water temperature was approximately 20 °C. Following complete cooling to room temperature, the samples underwent artificial aging by being immersed in an oil bath set at 121 °C for a period of 24 h. Additionally, samples of the same dimensions were drilled and embedded with thermocouples at different distances of 3 mm, 13 mm, 23 mm, 53 mm, 78 mm, and 98 mm from the water spray end. The cooling curves at these positions during the end-quenching process were recorded, and the following average cooling rates were obtained (in the 420–230 °C range) [2]: 1048 °C/min, 782 °C/min, 526 °C/min, 152 °C/min, 132 °C/min, and 129 °C/min, as shown in Figure 1.

### 2.2. Immersion Tests

Cut end-quenched and aged specimens (2 mm thick) were used for peel corrosion experiments following the GB/T 22639-2008 standard [27]. The solution had an area to volume ratio of 25 cm^2^/L, and the experimental temperature was maintained at 25 ± 1 °C. After 48 h of corrosion, the specimens were rated according to the standard. The solution system used was an EXCO solution (0.5 mol/L KNO_3_ + 0.1 mol/L HNO_3_ + 4 mol/L NaCl).

### 2.3. Microstructure Examination

Samples were taken from the corresponding positions of the specimens for microstructural analyses. The metallographic samples were subjected to coarse grinding, fine grinding, and polishing, followed by etching with the corrosion reagent (Graff Sargent). The composition of the reagent was 1% HF, 16% HNO_3_, 83% H_2_O (by volume), and 3 g of CrO_3_, which effectively distinguished between the unrecrystallized and recrystallized regions in the alloy. The microstructure was observed using an XJP-6A (T-Bota, Nanjing, China) metallographic microscope, and further microstructural observations and energy-dispersive spectroscopy (EDS) analysis were conducted using the Quanta-200 SEM (FEI, San Jose, CA, America). The grain structure analyses were performed using a EVOMA10 SEM (Carl Zeiss, Jena, Germany) equipped with an OXFORD EBSD detector. Tecnai G2 F20 transmission electron microscopy (TEM) was employed for observing quenching precipitation phases. Both the EBSD and TEM specimens were prepared using the dual-jet electropolishing method, with electrolytic thinning carried out in a mixture of 20% nitric acid solution and 80% methanol. The electrolyte temperature was controlled at approximately −25 °C using liquid nitrogen.

## 3. Results

### 3.1. Grain Structure

Figure 2 displays the EBSD images of the homogenized cast ingot and hot-rolled samples of the 7085 alloy. In Figure 2a, it can be observed that almost all the grains exhibit an equiaxed shape with non-uniform grain sizes, and the GBs are predominantly HAGBs. The as-cast samples with this grain structure are subsequently referred to as equiaxed grain (EG). Figure 2b reveals that the grains after hot rolling exhibit a distinct elongated shape aligned along the rolling direction. In Figure 2c,f, the recrystallized structure is depicted by the blue region, the sub-grain structure is depicted by the yellow region, and the deformed structure is depicted by the red region. The deformed sample contains both recrystallized grains and sub-grains. The black lines indicate HAGBs, while the white lines represent low-angle grain boundaries (LAGBs). Along the deformation direction, there are numerous LAGBs in the sub-grain and deformed regions, which are typical grain structures of 7XXX aluminum alloys after hot deformation and solid solution treatment. Hot-rolled samples with this grain structure are subsequently labeled bar grain (BG). Furthermore, compared to the EG sample, the BG sample obtained through rolling exhibits a significant reduction in grain size. The average size has decreased from the original 148 μm to 70 μm, resulting in BG samples having a greater abundance of GBs, as shown in Figure 2d,e.

### 3.2. Grain Boundary Quenching Precipitated Phase

The SEM images of EG and BG specimens at different cooling rates are shown in Figure 3. In EG and BG samples, coarse white second-phase particles were observed. EDS analysis revealed that the chemical composition of these particles (wt%) was as follows: Al: 55.24~71.26, Fe: 8.58~14.20, and Cu: 18.44~28.32, with traces of Zn. These particles are likely to be Al_7_Cu_2_Fe phases [28]. The formation of this iron-containing phase occurs during the solidification process and exhibits good thermal stability. Subsequent processes such as solution treatment and quenching have a small influence on the distribution and size of this phase. In Figure 3a, it can be observed that in the EG sample at a cooling rate of 1048 °C/min, only the presence of white primary phases is observed, and no quenching precipitates are observed. Figure 3b reveals that at a cooling rate of 129 °C/min, fine quenching precipitates are observed at HAGBs, and some precipitates can also be seen within the grains. Based on the EDX analysis results, the main constituents of this phase are Zn and Mg, along with a small amount of Cu, and it corresponds to the η phase [23]. From Figure 3c, it is evident that in the BG sample at a cooling rate of 1048 °C/min, only white primary phases are observed, and no η phases are observed. Figure 3d shows that at a cooling rate of 129 °C/min, there are evident η phases at the HAGBs, and η phases can also be observed at the LAGBs, although the size of the precipitates is smaller. During the slow quenching process, due to the higher interfacial energy of both HAGBs and SGBs, favorable conditions are created for the nucleation and growth of the η phase. As a result, the η phase primarily forms at HAGBs and SGBs [29,30]. Additionally, SGBs have lower energy compared to HAGBs, resulting in slower solute atom diffusion along SGBs. As a result, the subgrain boundary phases (SGBPs) exhibit smaller sizes.

Figure 4 represents the TEM images of EG and BG samples at different cooling rates. From Figure 4a, it can be seen that in the EG sample at a cooling rate of 1048 °C/min, the size of the η precipitates at HAGBs is small and uniform, approximately 24 nm, and the PFZ is narrow, approximately 26 nm. As the cooling rate declines to 129 °C/min, the GBPs’ size becomes larger, the distribution becomes discontinuous, and the PFZ becomes more pronounced. The η phase size and the PFZ width increase to approximately 64 nm and 86 nm, respectively, as shown in Figure 4b. This is because during the slow quenching process, vacancies diffuse and solute atoms to HAGBs, resulting in the formation of larger GBPs and a wider PFZ [31,32]. In Figure 4c, it can be observed that in the BG sample at a cooling rate of 1048 °C/min, which corresponds to a higher cooling rate, the precipitates are age precipitates that exhibit a continuous distribution with a size of approximately 14 nm, and the PFZ width is approximately 28 nm. However, the precipitation at the SGBs is not significant, as shown in Figure 4d. When the cooling rate reduces to 129 °C/min, the size of the GBPs increases, and the distance between the precipitates becomes wider. The PFZ also becomes very pronounced. The GBPs’ size and the PFZ width are approximately 114 nm and 110 nm, respectively, as shown in Figure 4e. Additionally, quench precipitates can also be observed at the SGB (Figure 4f), with smaller sizes compared to the precipitates at the HAGB.

Figure 5 shows the size of the η phase at GBs and the width of the PFZ at different cooling rates. As shown in Figure 5a, the GBPs’ size significantly augments as the cooling rate reduces, and this increase is more pronounced in the BG samples. At a cooling rate of 1048 °C/min, the GBPs’ (η phase) average size in the BG samples is approximately 10 nm smaller than in the EG samples. However, when the cooling rate drops to 129 °C/min, the GBPs in the BG samples are 50 nm larger than in the EG samples. This indicates that the hot deformation grain structure promotes an increase in the size of the GBPs at lower cooling rates. In Figure 5b, it is evident that the width of the PFZ at GBs increases as the cooling rate declines, and this rise is more apparent in the BG samples. At a cooling rate of 1048 °C/min, the PFZ width at GBs in the BG samples is 2 nm larger than in the EG samples. However, when the cooling rate drops to 129 °C/min, the PFZ width at GBs in the BG samples is 24 nm larger than in the EG samples. The hot-deformed grain structure promotes an increase in the PFZ width at GBs at lower cooling rates. This indicates that hot deformation facilitates the nucleation and growth of the η phase at GBs and widens the PFZ.

The composition of GBPs at different cooling rates can be seen in Figure 6. The observation reveals a considerably higher content of Zn and Mg elements in the GBPs compared to Cu. As the cooling rate decreases, the Zn, Mg, and Cu content in the GBPs rise, with a much larger increase in Zn and Mg content compared to Cu content. At the same cooling rate, the GBPs in the BG sample have a slightly higher content of Zn and Mg compared to the EG sample, while the Cu content is similar. When the cooling rate descends from 1048 °C/min to 129 °C/min, the content of Zn, Mg, and Cu in the GBPs of the EG sample rises from 3.2%, 2.3%, and 0.9% to 12.3%, 11.2%, and 2.1%, respectively. Similarly, the content of Zn, Mg, and Cu in the GBPs of the BG sample increases from 3.4%, 2.5%, and 1.1% to 14.4%, 12.9%, and 2.3%, respectively. This indicates that there is not a significant difference in the composition of GBPs between the two grain structures.

### 3.3. Exfoliation Corrosion Immersion Test Results

The SEM images of the sample with a cooling rate of 129 °C/min soaked in EXCO solution for 1 h are shown in Figure 7. The surface of the EG samples exhibits noticeable corrosion products, primarily distributed at GBs, as shown in Figure 7a. According to EDX analysis, the predominant corrosion product is Al_2_O_3_, as shown in Figure 7c. This is mainly due to the susceptibility of grain boundary quenching precipitates to corrosion. Since the grains are predominantly equiaxed, the corrosion follows the GBs, although the extent of exfoliation is not significant. The surface of the BG sample exhibits more severe corrosion, with abundant and larger-sized corrosion products. The presence of recrystallized grains with high aspect ratios and a large number of HAGBs in the BG sample leads to the generation of a substantial amount of corrosion products. This further exacerbates the occurrence of EXCO, indicating a higher sensitivity to EXCO, as shown in Figure 7b.

The TEM images of the sample with a cooling rate of 129 °C/min soaked in EXCO solution for 1 h can be seen in Figure 8. It can be observed from the figure that the precipitates at the GBs of both the EG and BG samples have been completely corroded. Previous studies have shown that the potential of the grain boundary η phase is −1.05 V, the potential of the PFZ is −0.85 V, and the potential of the matrix inside the grain is −0.75 V [26]. Therefore, in the microgalvanic cells formed within the corrosive medium, GBPs and PFZs act as the anodes and are preferentially corroded, leading to GBs becoming the initiation points and propagation paths for corrosion.

Figure 9 shows the corrosion morphology of the end-quenched samples at different soaking times in the EXCO solution. For the end-quenched samples of the EG alloy, uniform pitting corrosion is observed along the water spray direction. With increasing soaking time, slight “blistering” and “peeling” occur, but there is minimal corrosion product in the solution, and no significant evidence of EXCO is observed. According to the grading standard GB/T 22639-2008 for alloy EXCO, the corrosion rating at the low cooling rate after 48 h is classified as PC grade (Figure 10a). However, for the end-quenched samples of the BG alloy, as the cooling rate decreases, a larger number of bubbles are generated during the immersion process, indicating a more intense reaction with the corrosive solution. At the initial stage of immersion, a slight pitting corrosion appears on the alloy’s surface, which intensifies over time. After 8 h, blistering becomes evident, and after 24 h, with further reduction in cooling rate, the alloy exhibits more pronounced blistering, peeling, and severe delamination on the surface, penetrating deep into the metal. The areas with smaller cooling rates exhibit more corrosion products and more severe EXCO (Figure 9h). After 48 h of immersion (Figure 9j), larger areas of corrosion are observed at locations with lower cooling rates. The surface blistering has completely ruptured and delaminated, penetrating deep into the interior of the metal. The solution contains a significant amount of detached corrosion products, and the corrosion rating has reached the EB grade (Figure 10b). The corrosion ratings for the EG and BG samples are shown in Figure 10, indicating that corrosion becomes more severe with increasing immersion time, particularly at lower cooling rates (152–129 °C/min), but the difference in their ratings is not significant.

The cross-sectional corrosion morphologies of EG and BG specimens at different cooling rates are shown in Figure 11. The extent of corrosion in the EG samples increases slightly as the cooling rate decreases. Corrosion occurs along the GBs, and in some areas, entire grains are corroded, resulting in the formation of corrosion pits. At a cooling rate of 1048 °C/min, corrosion is observed along the GBs, and significant corrosion is also present within the grains, as shown in Figure 11a. At 129 °C/min, almost the entire grain is corroded, and the maximum depth of corrosion is approximately the size of a single grain, as shown in Figure 11b. Similarly, the degree of EXCO in the BG samples increases significantly as the cooling rate decreases. At a cooling rate of 1048 °C/min, the EXCO is observed, and the maximum depth of corrosion is similar to that in the EG samples (Figure 11c). As shown in Figure 11e,f, at 129 °C/min, pronounced EXCO occurred. Simultaneously, significant corrosion cracks were observed and propagated along specific paths. Figure 11h is the inverse pole figure of corrosion cracks, where the black regions represent corroded areas, the red lines represent HAGBs, and the white lines represent LAGBs. It was observed that there were evident corrosion cracks along the HAGBs, and some of the SGBs were also corroded. Therefore, it can be inferred that the corrosion cracks primarily propagate along the HAGBs.

Figure 12 illustrates the relationship between the corrosion depth and the cooling rate at different cooling rates. It can be observed that the corrosion depth of both samples increases with decreasing cooling rates. However, the variation in corrosion depth for the EG sample is relatively small. At a cooling rate of 1048 °C/min, the maximum and average corrosion depths of the BG sample are 30 μm and 28 μm greater than those of the EG sample, respectively. At a cooling rate of 526 °C/min, the maximum and average corrosion depths of the BG sample are 50 μm greater than those of the EG sample. At 129 °C/min, the maximum and average corrosion depths of the BG sample (315 μm and 267.5 μm, respectively) are 125 μm and 120 μm greater than those of the EG sample. This indicates that the peel corrosion of the BG sample becomes more severe at very low cooling rates. Additionally, the corrosion depth of the samples continuously increases with decreasing cooling rates. By performing linear regression analysis on the data, it is found that the average corrosion depth (H) has the following logarithmic relationship with the cooling rate (CR):H_EG_ = 313.7 − 78.8 lg(CR)(1)
H_BG_ = 587.8 − 160.7 lg(CR)(2)

The linear correlation coefficients are 0.996 and 0.994, respectively, indicating a strong linear relationship between the logarithm of the average corrosion depth and the cooling rate within the studied range.

It is generally believed that the occurrence of EXCO in high-strength aluminum alloys is related to the characteristics of grain boundaries and the precipitation of phases at grain boundaries and PFZs [26,33]. On the one hand, the corrosion products formed on the aluminum alloy in the corrosive medium create an outward driving force. This outward driving force is related to the shape of the grains, and the more the grains are elongated, the greater the outward force that is generated. EXCO follows the stress corrosion cracking mechanism, where the corrosion product wedges generate tensile stress concentration at the crack tip, leading to the propagation of corrosion through the stress corrosion cracking (SCC) mechanism. As long as the tensile stress exists at the corrosion front, the EXCO will continue to propagate. After immersing different grain structure samples in an EXCO solution for 48 h, different types of localized corrosion occurred, as shown in Figure 9. In the corrosive solution, extensive corrosion products are generated after the sample undergoes corrosion, creating wedging forces that result in the delamination of the upper layer of the metal. Meanwhile, high aspect ratio grains lead to the gradual accumulation of surface strain, eventually resulting in the formation of larger blistering, delamination, and exfoliation morphologies [11]. Compared to EG samples, BG samples possess a significant amount of high-aspect ratio grain structure, which is more conducive to the propagation of corrosion cracks, thus exhibiting a higher sensitivity to EXCO.

On the other hand, in 7XXX-series aluminum alloys, EXCO typically initiates at GBs and propagates along them into the material’s interior. This is attributed to the significant potential difference between GBPs and PFZs at GBs and the matrix. In the corrosive medium, GBPs and PFZs with lower potentials are preferentially dissolved as the anode, forming preferential pathways for anodic dissolution along GBs [34], resulting in corrosion cracks propagating along GBs, as illustrated in Figure 11. Based on the experimental results, the schematic diagrams of EXCO propagation for EG and BG samples under different cooling rates are shown in Figure 13. In EG samples, due to their smaller grain aspect ratio, corrosion propagates slower along HAGBs, resulting in lower EXCO sensitivity, as depicted in Figure 13a,b. In comparison to EG samples, BG samples exhibit reduced grain size after hot rolling, leading to a significant increase in the number of GBs, as shown in Figure 2. Additionally, hot rolling promotes the nucleation and growth of GBPs, forming wider PFZs. This provides more favorable conditions for the expansion of corrosion cracks, contributing to the higher EXCO sensitivity in BG samples.

Furthermore, the size, spacing, and chemical composition of GBPs and the width of PFZs significantly influence the alloy’s sensitivity to EXCO [35]. At lower cooling rates, GBPs continue to absorb surrounding atoms and grow, leading to larger PFZ widths. Microstructural observations reveal that as cooling rates decrease, both EG and BG samples exhibit increased GBPs and PFZ widths. As shown in Figure 13b,d, this makes grain boundaries more susceptible to corrosion and accelerates the propagation of corrosion cracks. Simultaneously, the content of Zn and Mg elements in GBPs is notably higher than that of Cu, and with decreasing cooling rates, the increase in Zn and Mg content far exceeds that of Cu (Figure 5). This results in a greater potential difference between GBPs and the matrix, further accelerating the corrosion crack propagation rate along grain boundaries [36]. The results demonstrate that as cooling rates decrease, the EXCO sensitivity of both EG and BG samples gradually increases.

Existing research has indicated that for 7XXX-series aluminum alloys, EXCO primarily extends along HAGBs, while propagation along SGBs is challenging [11,31]. This is attributed to the higher interfacial energy of HAGBs, which serve as the primary diffusion pathways for atoms, facilitating the nucleation and growth of GBPs through continuous atom absorption [36,37]. Consequently, there are more and larger GBPs at HAGBs, accompanied by wider PFZs. In contrast, SGBs possess lower interfacial energy, hindering the nucleation and growth of GBPs and resulting in narrower PFZs that are less conducive to forming independent anodic regions, making it difficult to establish corrosion pathways. However, as cooling rates decrease, GBPs at SGBs also grow larger, and PFZs widen, providing favorable conditions for the expansion of corrosion cracks. As depicted in Figure 13d, when the cooling rate decreases to 129 °C/min, corrosion preferentially extends along both HAGBs and SGBs, resulting in increased corrosion depth and a wavy distribution of macroscopic corrosion morphology. This further elucidates that decreasing cooling rates reduce the resistance of the alloy to EXCO.

## 4. Conclusions

For the 7085 alloy, BG samples exhibit higher sensitivity to EXCO compared to EG samples, regardless of the cooling rate. This is because BG samples have larger aspect ratio grains, which lead to the accumulation of surface stress during the corrosion process, resulting in faster crack propagation along the GBs. Additionally, BG samples have a higher number of GBs, which results in a greater amount of GBPs and a higher sensitivity to EXCO.With a decrease in cooling rate, both EG and BG samples show an increasing trend in EXCO sensitivity. This is attributed to the slower cooling rate, which leads to an increase in the size of GBPs and the width of the PFZ. Both the maximum corrosion depth and average corrosion depth significantly increase with decreasing cooling rates, with a higher rise observed in BG samples. In the meantime, a linear correlation can be established between the average depth of corrosion and the logarithm of the cooling rate.GBPs and their PFZ have a lower potential compared to the matrix, causing them to act as anodes and preferentially dissolve during corrosion. Consequently, corrosion cracks propagate along the GBs. Corrosion cracks in both EG and BG samples primarily propagate along HAGBs. At lower cooling rates, crack propagation along SGBs is observed in BG samples.

## Figures and Tables

**Figure 1 materials-16-05934-f001:**
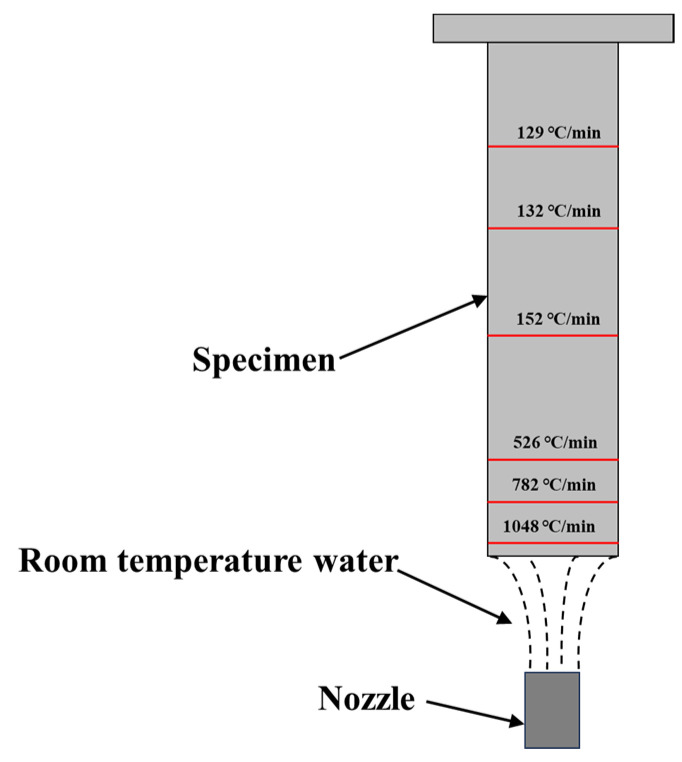
Schematic of end-quenching of the samples after solution heat treatment.

**Figure 2 materials-16-05934-f002:**
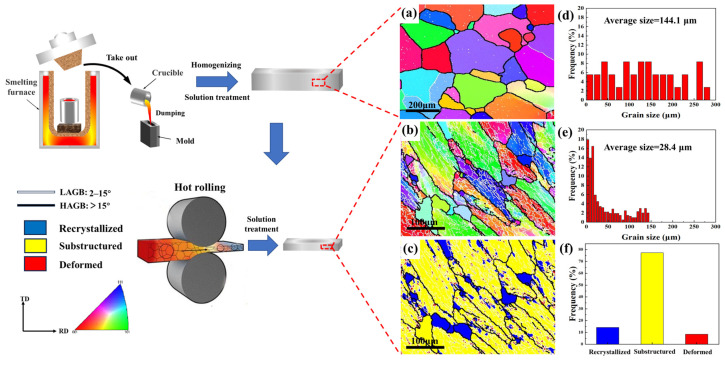
EBSD images of EG (**a**,**d**) and BG (**b**,**c**,**e**,**f**) samples: (**a**,**b**) IPF, (**d**,**e**) grain size, (**c**,**f**) DRX.

**Figure 3 materials-16-05934-f003:**
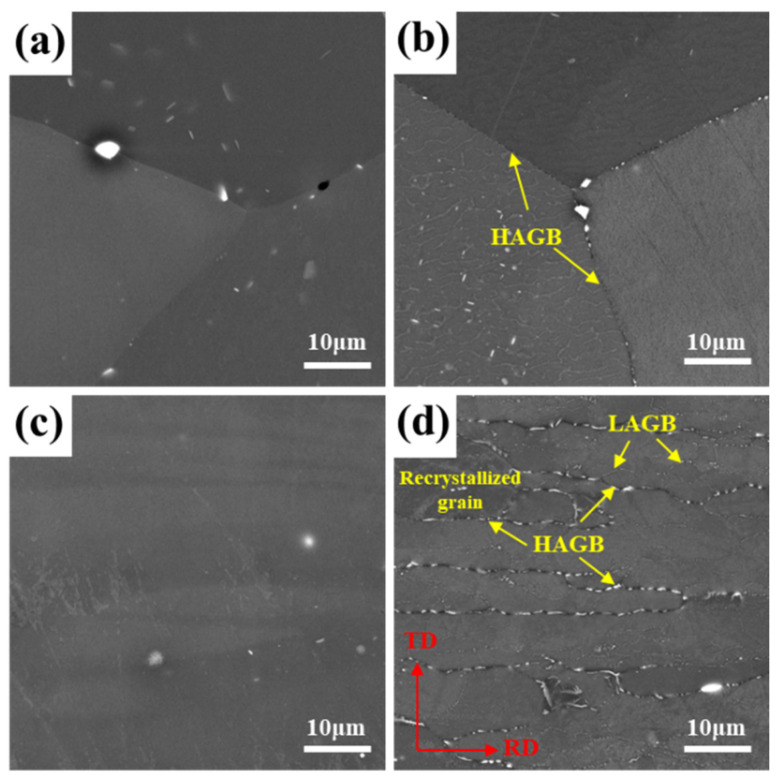
SEM images of EG (**a**,**b**) and BG (**c**,**d**) samples at different cooling rates: (**a**,**c**) 1048 °C/min, (**b**,**d**) 129 °C/min.

**Figure 4 materials-16-05934-f004:**
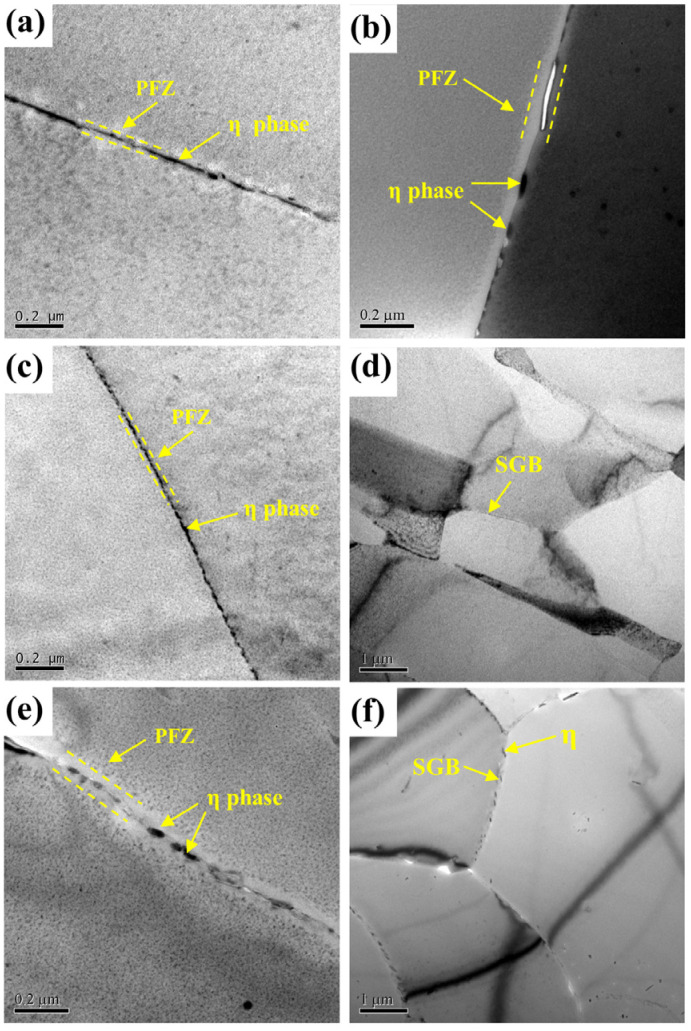
TEM images of EG (**a**,**b**) and BG (**c**–**f**) samples at different cooling rates: (**a**,**c**,**d**) 1048 °C/min, (**b**,**e**,**f**) 129 °C/min.

**Figure 5 materials-16-05934-f005:**
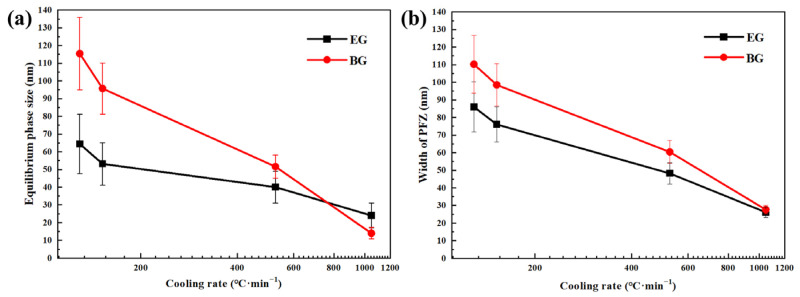
GBPs’ size and PFZ width at different cooling rates: (**a**) GBPs size, (**b**) PFZ width.

**Figure 6 materials-16-05934-f006:**
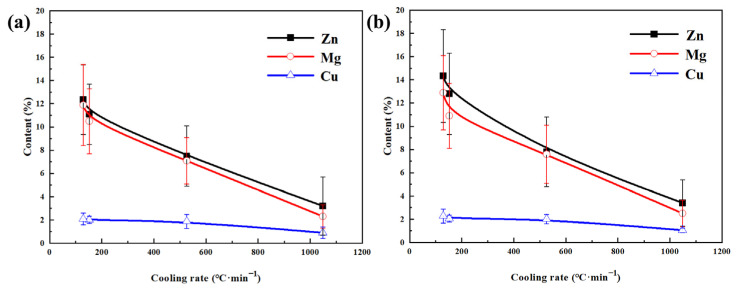
Compositions of GBPs at different cooling rates: (**a**) EG sample, (**b**) BG sample.

**Figure 7 materials-16-05934-f007:**
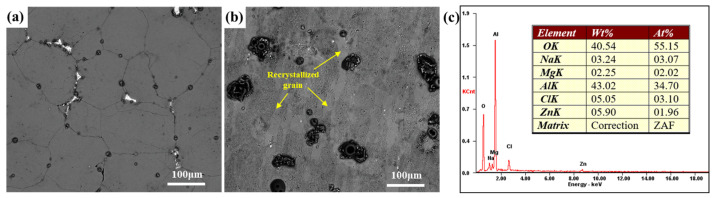
SEM image of the sample with a cooling rate of 129 °C/min soaked in EXCO solution for 1 h: (**a**) EG sample, (**b**) BG sample, (**c**) EDX.

**Figure 8 materials-16-05934-f008:**
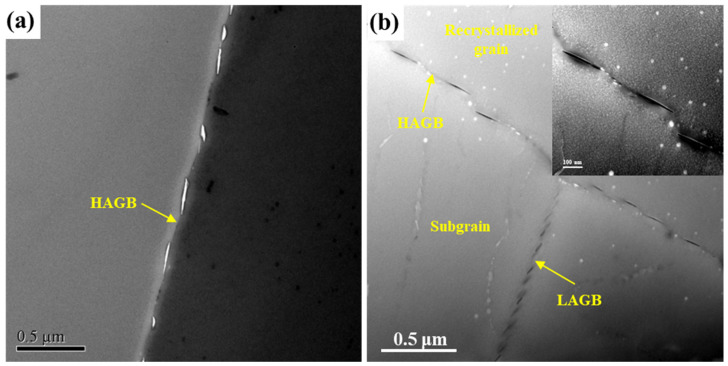
TEM image of the sample with a cooling rate of 129 °C/min soaked in EXCO solution for 1 h: (**a**) EG sample, (**b**) BG sample.

**Figure 9 materials-16-05934-f009:**
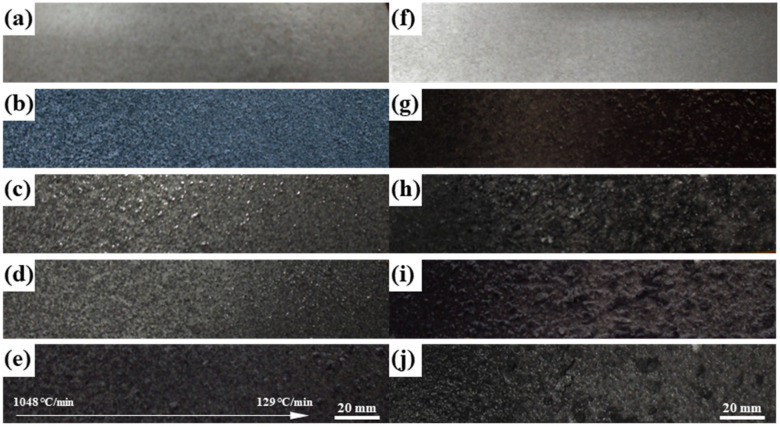
Corrosion morphology of the end-quenched samples at different soaking times in the EXCO solution, EG sample: (**a**) 2 h, (**b**) 12 h, (**c**) 24 h, (**d**) 36 h, (**e**) 48 h; BG sample: (**f**) 2 h, (**g**) 12 h, (**h**) 24 h, (**i**) 36 h, (**j**) 48 h.

**Figure 10 materials-16-05934-f010:**
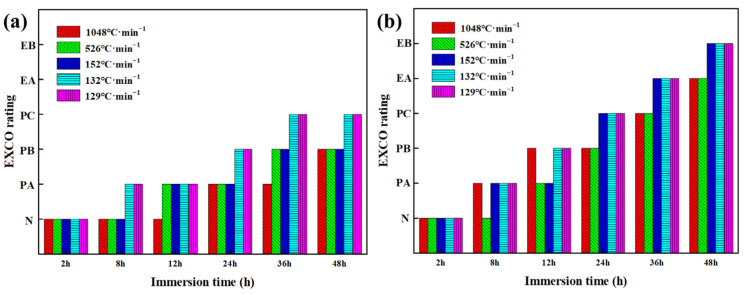
Corrosion ratings after soaking for different times: (**a**) EG sample, (**b**) BG sample.

**Figure 11 materials-16-05934-f011:**
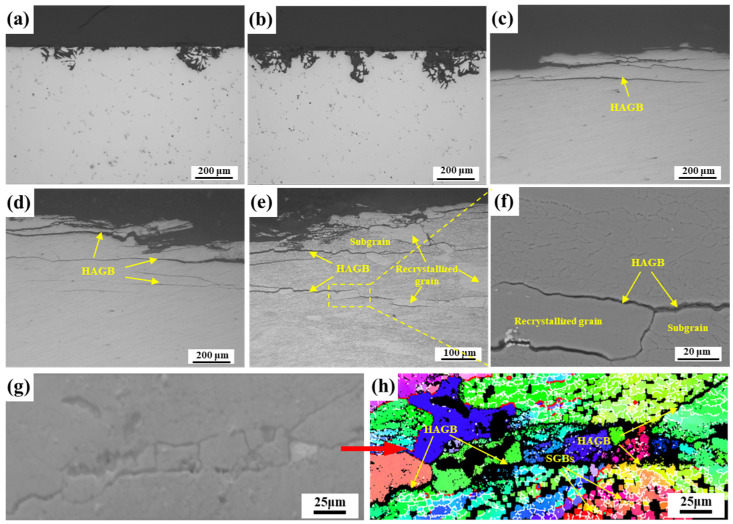
Cross-sectional corrosion morphologies of EG (**a**,**b**) and BG (**c**–**h**) samples at different cooling rates: (**a**,**c**) 1048 °C/min, (**b**,**d**,**e**–**h**) 129 °C/min.

**Figure 12 materials-16-05934-f012:**
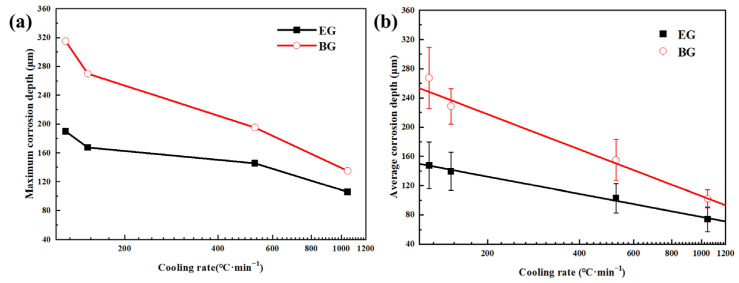
Relationship between the corrosion depth and the cooling rate at different cooling rates: (**a**) maximum corrosion depth, (**b**) average corrosion depth.

**Figure 13 materials-16-05934-f013:**
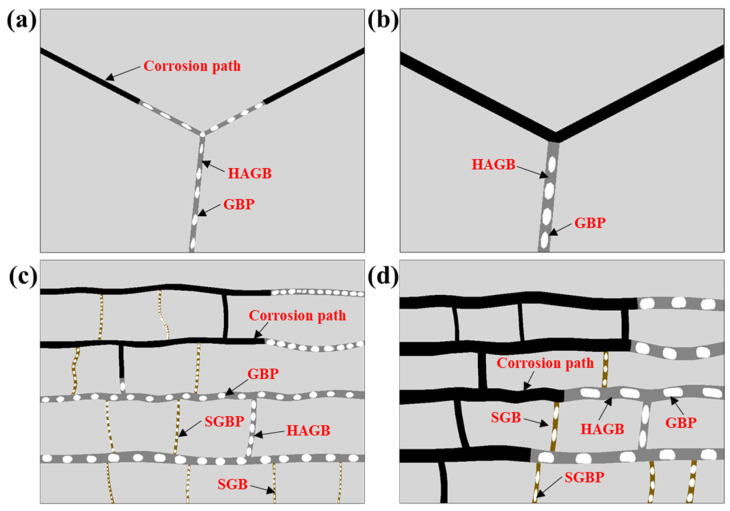
Schematic diagram of spalling corrosion expansion in EG (**a**,**b**) and BG (**c**,**d**) specimens at different cooling rates: (**a**,**c**) 1048 °C/min, (**b**,**d**) 129 °C/min.

**Table 1 materials-16-05934-t001:** Chemical compositions of the studied 7085 alloy (mass fraction, wt%).

Element	Zn	Mg	Cu	Zr	Fe	Si	Al
Content	7.5	1.6	1.7	0.11	<0.08	<0.06	Bal.

## Data Availability

Not applicable.

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
