# Peer review of "Effect of Grain Structure and Quenching Rate on the Susceptibility to Exfoliation Corrosion in 7085 Alloy"

_materials, 2023, doi:10.3390/ma16175934_

Round 1

Author Response

The replies to your suggestions are attached.

Reviewer 2 Report

In this article, the authors have studied the influence of grain structures and quenching rates on the exfoliation corrosion of 7085 Al alloy. The samples before and after corrosion have been characterized by various techniques. The manuscript can be accepted for publication with revisions based on following comments.

1.     The corrosion behavior of Al7085 are significantly affected by the grain size, grain boundary and internal stress. Can author provide the quantitative analysis of these samples instead of quantitative ones, for example, the average grain size, number of grain boundaries in different sample before corrosion experiment? These data can be correlated to the corrosion rate in the discussion.

2.     Line 188, the GBPs (η phase) size in the BG samples is 10 nm smaller than that in the EG samples. What is the error for the grain size? Is 10 nm within the error range? How was the size measured if they have irregular shapes?

3.     Lin215 to 225, what is meaning of the corrosion products?

4.     In figure 8, how do the authors confirm the TEM samples were prepared from the corrosion area, not from the unreacted part?

need to be corrected for some grammar error. 

Author Response

(The authors gave the same response as above.)

Reviewer 3 Report

the manuscript studies the effect of Grain Structure and Quenching Rate on the 2 Susceptibility to Exfoliation Corrosion in 7085 Alloy. The manuscript has serious flaws and cannot be considered for publication in its current form:

1- The language of the manuscript is not fit to the requirement of the journal. It should be revised by a native speaker who is an expert in the field.

2- The Introduction is not well organized and the results of the other references about the effect of grain structure were not discussed in detal. 

Please cite the followoing refernces about the effect of the grain structure on the corrosion behaviour of the 7xxx alloys and extend the introduction. 

https://doi.org/10.1016/j.ijhydene.2020.12.028

3- Regarding the initiation and propagation of the corrosion-induced crack, there is no enough explanation and discussion about the HAGB vs LAGB, etc. Please explain and clarify them carefully. 

4- The role of the PFZ was not explained in depth. What is the relationship between the HAGBs and PFZ with the craking behavior?

5- The discussion part is very poor and I'm not satisfied with the discussion and explanations. It should be extended with the focus on the explanation about the microstrucute and grain boundary orientaition with corrosion behavior.

The language of the manuscript is not fit to the requirement of the journal. It should be revised by a native speaker who is an expert in the field.

Author Response

(The authors gave the same response as above.)

Round 2

Author Response

Responses to the review suggestions are attached

Reviewer 3 Report

The comments were answered satisfactorily. It can be considered for publication. 

It required a minor revision in style and language.

Author Response

(The authors gave the same response as above.)
